# Amino-Acid-Substituted Perylene Diimide as the Organic Cathode Materials for Lithium-Ion Batteries

**DOI:** 10.3390/ma16020839

**Published:** 2023-01-15

**Authors:** Honggyu Seong, Wonbin Nam, Geongil Kim, Joon Ha Moon, Youngho Jin, Seung-Ryong Kwon, Joon-Hwa Lee, Jaewon Choi

**Affiliations:** Department of Chemistry and Research Institute of Natural Science, Gyeongsang National University, Jinju 52828, Republic of Korea

**Keywords:** perylene diimide, single molecule electrode, organic cathode, long-term cycling performance, lithium-ion batteries

## Abstract

One of the most effective cost reduction and green engineering projects is to introduce organic compounds to electrode materials instead of expensive inorganic-based materials. In this work, derivatives of perylene diimide substituted with amino acids (PDI_AAs) showed the characteristics of redox-active organic compounds and were, therefore, used as cathode materials of lithium-ion batteries (LIBs). Among the as-synthesized PDI_AAs, the L-alanine-substituted PDI (PDI_A) showed the most improved cycling performances of 86 mAhg^−1^ over 150 cycles with retention of 95% at 50 mAg^−1^. Furthermore, at a high current density of 500 mAg^−1^, PDI_A exhibited a long-term cycling performance of 47 mAhg^−1^ (retention to 98%) over 5000 cycles. In addition, ex situ attenuated total reflection Fourier-transform infrared spectroscopy (ATR FT-IR) analysis of electrodes at various charging states showed the mechanism of the charge-discharge process of PDI_A.

## 1. Introduction

Lithium-ion batteries (LIBs) are a class of energy storage systems in which the lithium-ion, as a charge carrier, can convert electrical energy into chemical energy through an electrochemical redox process [1]. Due to this powerful performance, LIBs have been widely used as power sources for various appliances, mobile products, and automobiles over the past three decades [2,3,4,5]. Therefore, many researchers have tried to increase the efficiency of LIBs and solve their safety problems. In material chemistry, inorganic materials such as LCO (lithium cobalt oxide), which is also known as LiCoO_2_, are commonly used as cathode materials in key components of LIBs’ systems [6,7]. However, there are limitations, such as environmental pollution from waste and the high prices of lithium and transition metals [8]. On the other hand, organic compounds can be used for LIBs at low cost, and organic materials are eco-friendly and sustainable compared with inorganic materials due to their component atoms, such as C, H, O, and N [9,10,11,12]. Due to these benefits, researchers are working on the development of organic compounds that can be used as cathode materials to replace inorganic materials [13,14,15].

Generally, the chemical structure of organic electrode materials is designed to have a high theoretical capacity [16]. Simultaneously, organic cathode materials should have redox activity for lithiation and delithiation in LIBs systems with organic electrolytes and certain additives. Like 1,4-benzoquinone (1,4-BQ) and pyromellitic diimide (PMDI), quinone and diimide derivatives with a redox-active carbonyl group have been reported as organic electrode materials that can be used for LIBs. Furthermore, a variety of compounds have been studied to commercialize organic cathodes [17,18,19,20]. However, small molecules usually undergo dissolution in organic electrolytes during charge–discharge cycling. Therefore, in order to find appropriate organic compounds as cathode materials, it is necessary to solve the problem of the dissolution of electrode materials during repeated charging and discharging processes, which results in reduced cycling performance. Recently, a new strategy to prevent dissolution using polymerization or molecular engineering has emerged [21,22].

Due to their highly stable aromatic core, perylene-based organic cathodes have been studied and developed using various strategies. Zhu et al. reported the lithium-ion storage ability of perylene diimide and its N-substituted derivatives [23]. Sharma et al. synthesized perylene-based polyimide and applied it as a cathode material for LIBs [24]. Recently, Medabalmi and Ramanujam synthesized glycine-substituted perylene diimide, which resulted in outstanding long-term cycling performances [25]. Inspired by these works, we chose perylene diimide substituted with three types of amino acids (PDI_AAs). The PDI_AAs were synthesized through the reaction of 3,4,9,10-perylene tetracarboxylic dianhydride (PTCDA) with L-amino acid (L-alanine (A), L-valine (V), and L-histidine (H)). As shown in Figure 1, these products were labelled as PDI_A, PDI_V, and PDI_H, respectively. The PDI_AAs cathode exhibited a high-rate capability of more than 90%. Among the as-synthesized PDI_AAs, PDI_A also showed excellent cycling performance and stability, with 98% retention over 5000 cycles at 500 mAg^−1^. Moreover, in this article, we discuss the lithiation structure of the PDI_A cathode through the ex situ ATR FT-IR spectrum and the DFT calculation of the discharged electrode.

## 2. Materials and Methods

### 2.1. Materials

3,4,9,10-Perylenetetracarboxylic dianhydride (PTCDA), imidazole, L-alanine, L-valine, L-histidine, and N-methyl-2-pyrrolidone (NMP) were purchased from the Sigma Aldrich Corp. Li metal foil was purchased from Alfa Aesar. Concentrated hydrochloric acid (HCl, 35–37%) was obtained from the Samchun Pure Chemical Co., Ltd. (Seoul, Republic of Korea). Carbon black (super P), a CR2032 coin cell assembly package, a 1 M lithium hexafluoro phosphate (LiPF_6_) solution in ethylene carbonate (EC) and diethyl carbonate (DEC) (EC:DEC = 1:1 volume ratio), and Al metal foil were purchased from the Wellcos Corp. (Gunpo-si, Republic of Korea). All the chemicals were used without further purification.

### 2.2. Characterization

Attenuated total reflection Fourier-transform infrared (ATR FT-IR) absorption spectra were obtained using a Thermo scientific US/IS5 FT-IR spectrometer (Waltham, MA, USA). ^1^H and ^13^C nuclear magnetic resonance (NMR) spectroscopy were conducted using a Bruker DRX300 300 MHz FT-NMR spectrometer (Billerica, MA, USA) in DMSO-*d*_6_ with chemical shift (δ) given in parts per million (ppm). Multiplicities were denoted as follows: s (singlet), d (doublet), t (triplet), m (multiplet), dd (doublet of doublets), and dt (doublet of triplets); the coupling constant (J) was given in hertz (Hz). The WonAtech ZIVE SP1 electrochemical workstation and WonAtech multichannel potentiostat (WBCS3000S) were used for the electrochemical experiment and the galvanostatic charge–discharge cycling test.

### 2.3. Synthesis of PDI_AAs

Under argon, imidazole and PTCDA (0.7846 g 1.0 eq, 2.0 mmol) were added to a flame-dried 50 mL two-neck Schlenk flask with a condenser. Then, the mixture was heated at 120 °C until it turned into a dark red solution. After the addition of the desired L-amino acid under argon, the reaction mixture was stirred at T (reaction temperature) for t (reaction time). After cooling to 90 °C, 20 mL of H_2_O was added under argon, and stirred for 2 h. To remove residual PTCDA, the reaction mixture was filtered with DI water. After Conc. HCl (aq.) was added to the solution until pH 1–2, the precipitate was obtained by suction–filtration, washed with DI water, then dried at 80 °C.

PDI_A: imidazole(10 g), L-alanine (0.3742 g, 2.1 eq, 4.2 mmol), T: 120 °C, t: 14 h; Dark brownish-red solid (0.6945 g, 65% yield); ^1^H NMR (300 MHz, DMSO-*d_6_*) δ 7.79 (t, J = 6.9 Hz, 4H), 7.63 (t, J = 9.8 Hz, 4H), 5.56–5.44 (m, 2H), 1.66 (d, J = 7.1 Hz, 6H). ^13^C NMR (75 MHz, DMSO-*d_6_*) δ 171.28 (carboxylic acid C=O), 161.48 (d, J = 2.8 Hz, imide C=O), 132.62, 130.11, 127.17, 123.87, 122.93, 121.29 (d, J = 3.3 Hz), 48.66 (-N-CH-), 14.45 (-CH_3_). ATR FT-IR cm^−1^ 1744.3 (imide asymmetrical C=O stretch) 1691.3 (imide symmetrical C=O stretch) 1636.8 (carboxylic acid C=O stretch) 1588.1 1572.7 1507.1 1483.0 1436.7 1401.0 (perylene ring moieties) 1458.4 1363.9 (C-H bending) 1337.4 (imide C-N stretch) 1251.6 (carboxylic C-O stretch) 1176.8 (amino acid C-N stretch).

PDI_V: imidazole (10 g), L-valine (0.4920 g, 2.1 eq, 4.2 mmol), T: 120 °C, t: 14 h; Dark red solid (1.0080 g, 85% yield); ^1^H NMR (300 MHz, DMSO-*d_6_*) δ 8.34–8.06 (m, 8H), 5.19 (dd, J = 9.2, 2.3 Hz, 2H), 2.74 (dt, J = 9.1, 6.6 Hz, 2H), 1.29 (d, J = 6.4 Hz, 6H), 0.82 (d, J = 7.0 Hz, 6H). ^13^C NMR (75 MHz, DMSO-*d_6_*) δ 170.66 (carboxylic acid C=O), 162.28 (d, J = 4.7 Hz, imide C=O), 133.29 (d, J = 12.2 Hz), 130.95, 127.90, 124.58, 123.17 (d, J = 13.3 Hz), 121.38 (d, J = 8.8 Hz), 58.12 (-N-CH-), 27.09 (-CH-), 22.29 (-CH_3_), 19.07 (-CH_3_). ATR FT-IR cm^−1^ 1743.3 (imide asymmetrical C=O stretch) 1694.6 (imide symmetrical C=O stretch) 1634.9 (carboxylic acid C=O stretch) 1590.5 1573.6 1506.1 1482.0 1435.3 1402.5 (perylene ring moieties) 1470.9 1462.7 1365.8 (C-H bending) 1337.4 (imide C-N stretch) 1250.6 (carboxylic C-O stretch) 1173.5 (amino acid C-N stretch).

PDI_H: imidazole (2 g), L-histidine (0.9309 g, 3.0 eq, 6.0 mmol), T: 140 °C, t: 21 h; Dark purplish-red solid (0.9737 g, 73% yield); A drop of D_2_SO_4_ was added to increase solubility in DMSO-*d_6_* solvent for the NMR analysis. ^1^H NMR (300 MHz, DMSO-*d_6_*) δ 8.86 (d, J = 1.5 Hz, 2H), 8.73–8.23 (m, 8H), 7.38 (s, 2H), 5.79 (dd, J = 9.5, 4.6 Hz, 2H), 3.64 (d, J = 11.7 Hz, 2H), 3.38 (dd, J = 15.0, 9.4 Hz, 2H). ^13^C NMR (75 MHz, DMSO-*d_6_*) δ 170.33 (carboxylic acid C=O), 163.11 (imide C=O), 134.92, 134.53, 132.29, 130.37, 129.15, 126.16, 124.95, 122.58, 117.76, 53.26 (-N-CH-), 24.59 (-CH_2_-). ATR FT-IR cm^−1^ 1735.1 (imide asymmetrical C=O stretch) 1693.2 (imide symmetrical C=O stretch) 1650.8 (carboxylic acid C=O stretch) 1590.5 1570.1 1507.1 1482.5 1434.8 1401.5 (perylene ring moieties) 1468.0 1460.3 1358.1 (C-H bending) 1340.8 (imide C-N stretch) 1252.1 (carboxylic C-O stretch) 1172.0 (amino acid C-N stretch).

### 2.4. Assembly of Coin Cells

For the electrochemical measurement, 55 mg of PDI_AAs (PDI_A or V or H), 35 mg of Super P carbon black, and 10 mg of polyvinylidene fluoride (PVDF) were ground with NMP until it became homogeneous slurry. After the slurry was loaded onto Al foil with a doctor blade, the electrode was dried at 80 °C for one hour under air and for three hours under vacuum. After that, the electrode punched into a 10 mm diametric disk was fabricated into CR2032 coin-type cells with Li metal foil as the counter and reference electrode, 1 M LiPF_6_ in EC:DEC = 1:1 (*v*/*v*) solution as the electrolyte, and Celgard 3501 as a separator in a glove box filled with pure argon.

### 2.5. Computational Calculation

Geometrical optimizations for isolated molecules of PDI_A, PDI_V, and PDI_H in the vacuum state were performed at the B3LYP/6-31+G(d,p) level using the Gaussian 16W software package [26].

## 3. Results and Discussion

In this work, we synthesized derivatives of perylene diimide substituted with three types of L-amino acid (PDI_AAs) through the reaction of anhydride with amine, which is one of the well-known synthetic pathways used in the literature. (The detailed procedures are described in the experimental section) [27,28,29]. The chemical structures of the as-synthesized PDI_AAs series (PDI_A, PDI_V, and PDI_H) are shown in Figure 1 and confirmed by ^1^H NMR and ^13^C NMR (Appendix A). In addition, the ATR FT-IR absorption spectrum of the as-synthesized PDI_AAs shows the typical bands corresponding to imide C=O vibrations (symmetric and asymmetric stretch), carboxylic acid C=O stretch, and imide C-N stretch in Appendix A.

The lithiation–delithiation process of the PDI_AAs is shown in the cyclic voltammograms obtained between 1.0 V to 4.0 V vs. Li/Li^+^ of the potential window at a scan rate of 10 mVs^−1^ (Figure 1a,c,e). The cathodic–anodic potential in the first cycle of the PDI_AAs was observed at (PDI_A: 1.82/2.78 V), (PDI_V: 1.86/2.89 V), and (PDI_H: 1.73/2.94 V). As cyclic voltammetry (CV) was conducted, the cathodic–anodic potential of PDI_AAs was shifted to (PDI_A: 2.13/2.78 V), (PDI_V: 2.07/2.99 V), and (PDI_H: 1.75/3.22 V) at the 4th CV curves. These cathodic and anodic peaks indicated that the PDI_AAs electrodes were delithiated and lithiated during the CV sweep during the charge–discharge process on the electrode surfaces. In the case of PDI_A, the current density and redox activity were slightly enhanced during the cycle, which was due to the “activation” mentioned later. Except for PDI_A, the main redox-active pair of PDI_V and PDI_H was polarized over the cycle number. It was assumed that PDI_V and PDI_H were dissolved into electrolytes and became impurities, disrupting the electronic or ionic transfer. Furthermore, PDI_A and PDI_V showed small cathodic–anodic peaks after the 1st cycle. These peaks were (PDI_A: 2.80/3.16 V) and (PDI_V: 2.71/3.29 and 3.70 V). These results are analogous to the ones in a previous work [25]. Therefore, the CV curves indicate the lithiation–delithiation of PDI_A and PDI_V proceed through the two-electron transfer two-step process.

Based on the electrochemical behaviors of the PDI_AAs cathode in the CV curves, the constant current charge–discharge cycling test was conducted between 1.0–4.0 V vs. Li/Li^+^ at various current densities. The 1st, 10th, 50th, 100th, and 150th galvanostatic charge–discharge profiles of the PDI_AAs at a current density of 50 mAg^−1^ are depicted in Figure 1b,d,f. Each charge–discharge capacity of the PDI_AAs is (PDI_A: 64/91, 83/87, 103/102, 103/101, and 87/86 mAhg^−1^), (PDI_V: 106/168, 88/90, 80/79, 71/70 and 60/60 mAhg^−1^), and (PDI_H: 66/75, 63/65, 62/61, 52/52, and 39/38 mAhg^−1^). Notably, PDI_A, V, and H showed a plateau at about 2.60 V during the discharge process and around 2.35 V during the charge process, which corresponded to the initial points of the lithiation–delithiation in the CV curves.

Moreover, the reduction potential for lithiation of organic compounds was related to their lowest unoccupied molecular orbital (LUMO) level [30,31]. Therefore, the more suitable cathode materials for LIBs should have low LUMO levels. According to the cathodic peaks of the CV curve in this work, this suggested that PDI_A, which exhibited the highest discharge potential, had the lowest LUMO level among the as-synthesized PDI_AAs, as shown in Figure 2 and Table 1.

The cycling performances of the PDI_AAs cathodes showed specific discharge capacities at current densities of 50 mAg^−1^ and 500 mAg^−1^, as shown in Figure 3a,b. At a current density of 50 mAg^−1^, the first discharge capacities of the PDI_AAs were (PDI_A: 91 mAhg^−1^), (PDI_V: 169 mAhg^−1^), and (PDI_H: 75 mAhg^−1^). After 150 cycles, the specific discharge capacities and the retention-to-first-discharge capacities of the PDI_AAs were (PDI_A: 86 mAhg^−1^, 95%), (PDI_V: 60 mAhg^−1^, 36%), and (PDI_H: 38 mAhg^−1^, 51%), respectively. Moreover, as shown in Figure 3b, at a current density of 500 mAg^−1^, the first-5000th discharge capacities/retention of the PDI_AAs cathode were (PDI_A: 48/47 mAhg^−1^/98%), (PDI_V: 114/17 mAhg^−1^/15%), and (PDI_H: 60/6 mAhg^−1^/10%), respectively. In addition, the first coulombic efficiencies (CE) of the PDI_AAs at 50/500 mAg^−1^ were (PDI_A: 87/68%), (PDI_V: 63/75%) and (PDI_H: 71/75%), which were 100% close after several initial cycles, as shown in Appendix A. Remarkably, the cycling performance of the PDI_AAs was similar to the features of the CV curves. In the cases of PDI_V and PDI_H, they showed decreases in specific capacities during the cycling test, which indicates that they were dissolved into the electrolytes in the repeated charge–discharge process. Indeed, PDI_V and PDI_H are relatively more soluble in electrolyte than PDI_A as shown in Appendix A. Only PDI_A is not dissolved and can last in the organic electrolyte system. Generally, organic materials have poor kinetics, causing the disruption of electronic and ionic transfer. This problem can be overcome through an activation step during several initial cycles by the penetration of electrolyte into the electrode [32]. Therefore, we believe that it causes PDI_A to go through the activation step, resulting in an increase in specific capacity.

A comparison with the cycling performance of perylene-diimide-based single molecules as the organic cathode is shown in Table 2. Even if the specific capacity of PDI_A is lower than that of the others, PDI_A showed outstanding long-term cycling performance and stability, with 47 mAhg^−1^ over 5000 cycles at 500 mAg^−1^ and a high retention of 98%.

Figure 3c depicts the rate capabilities of PDI_AAs cathodes in each 15 charge–discharge cycle by increasing the discharge current density from 50 mAg^−1^ to 1000 mAg^−1^ and then reversing the decrease back to 50 mAg^−1^. Each average discharge capacity at various current densities (50, 100, 200, 500, 1000, 500, 200, 100, and 50 mAg^−1^) was (PDI_A: 83, 80, 72, 62, 55, 60, 70, 75, and 78 mAhg^−1^), (PDI_V: 73, 63, 55, 46, 40, 47, 55, 62, and 69 mAhg^−1^) and (PDI_H: 63, 57, 50, 37, 30, 38, 51, 56, and 62 mAhg^−1^). Notably, the recovery of the PDI_AAs cathodes was (PDI_A: 94%), (PDI_V: 95%), and (PDI_H: 98%), which showed their excellent rate capability above 90%.

The electrochemical impedance spectroscopy (EIS) was conducted in the frequency range of 1 MHz to 1 Hz. The fitted Nyquist plot and the equivalent circuit of PDI_A, V, and H from the EIS test are depicted in Figure 3d. The fitted charge transfer resistance (R_ct_) of the PDI_AAs cathode was (PDI_A: 338 Ω), (PDI_V: 376 Ω), and (PDI_H: 394 Ω), all of which were similar (Table 3). To further investigate the conduction property of the PDI_AAs cathode, we calculated the diffusion coefficient of Li-ion (D_Li_) with EIS data obtained through the previous reports [37,38]. The diffusion coefficient can be calculated from the diffusion length (l_D_) and the diffusion time constant (τ_D_), according to Equation (1):(1)$$D_{\mathrm{Li}}=\frac{l_{D}{}^{2}}{\tau_{D}}$$

The l_D_ is induced by Equations (2) and (3), as follows:(2)$$l_{D}=\sqrt{\tau_{s}\times D_{m}}=\sqrt{\tau_{s}\times \frac{\lambda_{D}{}^{2}}{\tau_{1}}}$$
(3)$$\lambda_{D}=\frac{d}{\delta }=\frac{d}{{\left(2\tan{\left(\Phi \right)}_{\max}\right)}^{2}}$$
where D_m_ is the diffusion coefficient of net mobility, λ_D_ is the Debye length of electrode, d is the half thickness of the electrode, δ is a dimensionless number, τ_s_ is the lifetime constant, τ_1_ is the relaxation time constant, and τ_s_ is the inverse of frequency at a point with the highest value in the Bode plot. Meanwhile, τ_1_ is the inverse of frequency at a point with the lowest −Z_im_ in the Nyquist plot. Also, τ_s_, τ_1_ and tan(Φ)_max_ can be obtained from the Bode plot and the loss tangent plot. In Equation (4), the τ_d_ can be calculated by the squared W_sc_. The W_sc_, a part of the Warburg O-element, was obtained from the equivalent circuit. The further detailed parameters are attached in the Appendix A.
(4)$$\tau_{D}=W_{\mathrm{sc}}{}^{2}$$

The calculated Li-ion diffusion coefficients of the PDI_AAs cathodes are (PDI_A: 4.55 × 10^−11^), (PDI_V: 8.18 × 10^−9^), and (PDI_H: 8.32 × 10^−12^), respectively (Table 3). The as-calculated diffusion coefficient indicated that the PDI_V cathode has the best ionic conductivity. We think this is related to the unpredictable initial specific capacity over its theoretical capacity of the faradaic process for two lithium ions. Meanwhile, even if PDI_A has relatively poor ionic conductivity compared to PDI_V, it showed the most powerful performances, due to its good electronic conductivity. Accordingly, this is not a simple problem to interpret in relation to the cycling performances, probably due to complexities such as the surface properties of the electrode.

To study the mechanism of lithiation–delithiation, the PDI_A cathode at different charging levels in the initial charge–discharge cycle was analyzed by ex situ ATR FT-IR in Figure 4a: pristine electrode (red) discharged to 1.00 V (green) and charged to 4.00 V (blue). As shown in Figure 4a, the absorption bands of 1748.2 cm^−1^ corresponding to the asymmetrical stretch of imide C=O disappeared at the electrode discharged to 1.00 V, then appeared again at the fully charged level. In the case of the absorption band of 1685.5 cm^−1^ corresponding to the symmetrical stretch of imide C=O, although it did not disappear completely, its trend was similar to one of asymmetrical stretch. This is a predictable phenomenon, since it is well-known that the carbonyl group of perylene diimide is involved in the lithiation process. Simultaneously, the absorption band of 1637.3 cm^−1^ corresponding to carboxylic C=O slightly decreased, then was restored during the discharge-charge process. In conclusion, all of the absorption bands related to C=O bonding of PDI_A showed disappearance and appearance during the discharge–charge process, whether these changes were clear or less clear. Interestingly, at 1606.4 cm^−1^, an unknown band was observed and showed opposite features for chemical materials with the carbonyl group. A comparison of spectroscopic information of different states (pristine, discharged, charged) is described in the Appendix A. Furthermore, as shown in Figure 4b, the optimized structure of mono- and di-lithiated PDI_A, based on DFT calculation, shows the lithium enolate coordinated with carboxylic acid, which is a seven-membered lithium metallacycle. These results indicate that the coordination of carboxylic acid C=O with lithium-ion occurs when the imide (-C=O) was reduced to the lithium enolate (=C-O-Li). In addition, we consider that this process proceeds reversibly through a two-electron two-step transfer, according to the CV curves (Figure 2a). Therefore, we believe that the carboxylic acid C=O of PDI_A is involved in the lithiation process and coordinates Li-ion of the lithium enolate through our proposed mechanism (Figure 2).

## 4. Conclusions

In this work, perylene diimide derivatives substituted with L-amino acids (PDI_AAs) were synthesized. The as-synthesized PDI_AAs were analyzed by NMR and ATR FT-IR, which exhibited chemical structure information. In order to apply PDI_AAs as an organic cathode for LIBs, we fabricated them into CR2032 coin-type cells and evaluated them by electrochemical analysis and galvanostatic charge-discharge tests. The PDI_AAs cathode showed an outstanding rate capability of more than 90%. Among the alternatives, the PDI_A cathode exhibited the highest cycling performances, with 86 mAhg^−1^ over 150 cycles at 50 mAg^−1^ with 95% retention, and a practical-capacity-to-theoretical-capacity ratio of 86% (100 mAhg^−1^). Furthermore, we conducted ex situ ATR FT-IR analysis and DFT calculations, and we proposed the mechanism of the lithiation–delithiation process of PDI_A, in which there was a lithium enolate formed by coordination with the carboxylic acid during the discharge process. We expect that these results will help us understand the electrochemical behaviors of the PDI_AAs series and develop a next-generation organic cathode with high-performance.

## Data Availability

The data presented in this study are available on request from the corresponding author.

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
