# Peer review of "Amino-Acid-Substituted Perylene Diimide as the Organic Cathode Materials for Lithium-Ion Batteries"

_materials, 2023, doi:10.3390/ma16020839_

Round 1

Reviewer 1 Report

Title: Amino acid substituted perylene diimide as the organic cathode materials for lithium-ion batteries

The work by Honggyu Seong et al. reports a modified amino acid substituted perylene diimide to construct the organic cathode materials for high-performance lithium-ion batteries. The as-synthesized PDI_AAs could be increased active sites and improved stability for the electrode. Meanwhile, the conjugated structure could be accelerated the charges (electrons and ions) delivery, which shows a better electrochemical performance. The reviewer would recommend a minor revision, because the following questions need to be addressed before publication.

1. The authors claim that “Geometrical optimizations for all molecules in vacuum state were performed at the b3lyp/6-31+g** level.” in “computational calculation” section. However, diffusion function is meaningless for these organic molecules including sample elements (like C, H, O, N). Furthermore, the basis set should improve to 6-311g* level to calculate the single points energy, and the calculation should consider solvation effect for whole process (like SMD).

2. How about the initial Coulombic efficiency for these electrodes?

3. The authors showed the result of the electrochemical impedance spectroscopy. However, these electrodes have similar charge transfer resistance. Whether the ion transfer resistance could distinguish these electrodes? Please reference these articles: Nano-Micro Letters 2022, 14, 50; Advanced Energy Materials 2021, 11 (16), 2100448.

4. The FT-IR spectra of PDI_A cathode at different charged state on the specific point exhibit imperceptible changes. The author should further to explain the results of the FT-IR spectra.

5. As show in Fig 4b, the calculation results of the lithiation process should consider solvation effect and identify whether there is the imaginary frequency for initial and final state.

6. The author should further explain the mechanism of the lithiation/delithiation for these organic molecules.

Reviewer 2 Report

The manuscript, "Amino acid substituted perylene diimide as the organic cathode materials for lithium-ion batteries" presents an investigation on three derivatives of amino acid substituted perylene diimide (PDI_AAs), as the cathode materials for lithium-ion batteries. The PDI_A displayed an improved cycling performances of 86 mAhg-1 over 150 cycles with a retention of 95 % at 50 16 mAg-1. The mechanism of charge-discharge process of the PDI_A is investigated by analyzing electrodes at different charging levels using ex-situ attenuated total reflection infra-red absorption spectroscopy (ATR FT-IR), making this manuscript suitable for publishing in the materials. However, prior to its acceptance for publication, the authors should address the following concerns:

1.     In page 9: The authors proposed that carboxylic C=O can coordinate with lithium-ion, so a carboxylic acid of PDI_A was involved in the lithiation process together when imide (-C=O) was reduced to lithium enolate (=CO-Li). According to the literature, only the imide carbonyl groups are involved in the lithiation and de-lithiation process, as the carbonyl group of carboxylate group is redox active only below 1 V vs. Li/Li+. Therefore, apart from ex-situ ATR FT-IR, can authors support their statement from the cyclic voltammetry (CV) curves of Figure 1?

Ref.: Electrochimica Acta 2017, 232, 244-253.

2.     In page 8, line no. 247: ‘…... absorption band corresponding to imide C=O at around 1748.2 cm-1 and disappeared during the discharge ……’, the authors should check for missing information.

3.     Any further typos or grammatical errors in the manuscript should be addressed by the authors.

Reviewer 3 Report

The manuscript discusses PDI_AA organic compounds and their application as economical and green cathode materials. Material synthesis, electrochemical performances, mechanisms and so on are included. It might be even better if the authors could consider the following comments:

1. In the description of the synthesis for three PDI_AAs, the high-level processes look similar while there are differences in the actual values of parameters. Would it be possible to integrate the common process along with a list that shows different parameter values for each compound?

2. In the discussion of electrochemical performances for using PDI_AAs as cathode materials, it looks like the capacity of the cell sample with PDI_A increases as it advances from the 1st cycle to the 10th cycle, while the other two do not undergo similar processes. Could you provide some explanation for this phenomenon?

3. For the mechanism for lithination/delithination, have the authors perform similar analysis to PDI_V and PDI_H like the one for PDI_A? I feel curious as all of them have carboxylic C=O bonds. Do the additional functional groups in PDI_H and PDI_V make some differences compared to PDI_A?

Reviewer 4 Report

The manuscript is well written and overall of a good quality. In the results/discussions section a comparison (e.g. table) with the state of the art in the field of organic cathodes is missing, thus it is difficult for a reader to understand the significance of the results.
